# Risk Assessment of Fenpropathrin: Cause Hepatotoxicity and Nephrotoxicity in Common Carp (*Cyprinus carpio* L.)

**DOI:** 10.3390/ijms25189822

**Published:** 2024-09-11

**Authors:** Gongming Zhu, Zhihui Liu, Hao Wang, Shaoyu Mou, Yuanyuan Li, Junguo Ma, Xiaoyu Li

**Affiliations:** 1State Key Laboratory of Antiviral Drugs, College of Life Science, Henan Normal University, Xinxiang 453007, China; 2018208@htu.edu.cn (G.Z.); liuzh2958@163.com (Z.L.); 15516748299@163.com (H.W.); 18439513168@163.com (S.M.); 15993030397@126.com (Y.L.); 041035@htu.edu.cn (X.L.); 2Henan International Joint Laboratory of Aquatic Toxicology and Health Protection, Henan Normal University, Xinxiang 453007, China; 3Pingyuan Laboratory, Xinxiang 453007, China

**Keywords:** fenpropathrin, common carp, liver, kidney, negative impacts, mechanisms

## Abstract

The synthetic pyrethroid pesticide fenpropathrin (FEN) is extensively used worldwide and has frequently been detected in biota and the environment, whilst the negative effects and toxicological mechanisms of FEN on non-target organisms are still unknown. In the present study, healthy immature common carp were treated with FEN (0.45 and 1.35 μg/L) for a duration of 14 days, and the negative impacts and possible mechanisms of FEN on fish were investigated. Biochemical analyses results showed that FEN exposure altered the levels of glucose (GLU), total cholesterol (T-CHO), triglyceride (TG), albumin (ALB), alkaline phosphatase (ALP), alanine transaminase (ALT), and aspartate transaminase (AST) in carp serum, and caused histological injury of the liver and kidney, indicating that FEN may cause hepatotoxicity and nephrotoxicity in carp. In addition, FEN also altered the activities of superoxide dismutase (SOD) and catalase (CAT) in carp serum, upregulated the levels of reactive oxygen species (ROS), and elevated the levels of malondialdehyde (MDA) in the liver and kidney. Meanwhile, tumor necrosis factor-α (TNF-α) and interleukin-1β (IL-1β) levels were also upregulated, indicating that oxidative stress and inflammatory reaction may be involved in the hepatotoxicity and nephrotoxicity caused by FEN. Furthermore, RNA-seq analysis results revealed that FEN treatment induced a diverse array of transcriptional changes in the liver and kidney and downregulated differentially expressed genes (DEGs) were concentrated in multiple pathways, especially cell cycle and DNA replication, suggesting that FEN may induce cell cycle arrest of hepatocytes and renal cells, subsequently inducing hepatotoxicity and nephrotoxicity. Overall, the present study enhances our comprehension of the toxic effects of FEN and provides empirical evidence to support the risk assessment of FEN for non-target organisms.

## 1. Introduction

Hazards posed by heavy and repetitive usage of pesticides have attracted worldwide attention. Pyrethroid insecticides have been widely utilized worldwide [1]. Fenpropathrin (FEN) is a kind of new synthetic, high efficiency, broad spectrum, and low cost pyrethroid acaricide and insecticide that is extensively applied in fruits, vegetables, cotton, cereals, and other crops [2]. The primary insecticidal mechanism of FEN involves disrupting voltage-gated sodium and chloride channels in insects, which impairs nerve function, leading to lethargy, paralysis, and ultimately mortality [3]. The presence of an α-cyano group in FEN also significantly amplifies its acute neurotoxicity [4]. It has been thought that FEN is less harmful to both non-target organisms and natural environments than the other insecticides. However, due to large-scale and long-term use of this pesticide, non-target organisms such as animals and even humans are exposed to FEN; therefore, there is much scope for the enhanced risk of intoxication in non-target organisms via contaminated water, vegetables, fruits, and other agents.

The presence of FEN has been frequently discovered in various biota and environmental samples. The detection rate of FEN in the collected fruit and vegetable samples reached as high as 2.3% [5], while FEN was also found in pollen and beebread collected throughout China [6]. The concentration of FEN in cropland soils within the Yangtze River Delta region, China, can reach a maximum level of 37.6 ng/g [7], and it has been found in the tested sediment samples collected from Liaohe River, Pearl River Estuary, and Taihu Lake in China [8,9]. Additionally, FEN residue has been frequently detected in various water sources; for instance, the maximum concentration of FEN in drinking water source areas was found to be 40 μg/L and 10–20 μg/L in Guiyang, China [10], and a recorded concentration of 0.15 μg/L was observed in the central lake of Qianwei Village, Chongming Island, China [11]. In the lake of Quaroun and Litani river, the recorded maximum concentration was up to 220.1 ng/L [12]. It has been reported that FEN exhibits persistent effects in aquatic environments; therefore, the residual FEN in the aquatic environment may inevitably pose a threat to the health of non-target organisms, especially aquatic organisms [13,14,15,16].

Fish are important aquatic organisms and exhibit a high degree of sensitivity to changes in the aquatic environment. They are frequently used as important bio-indicator species for assessing aquatic pollution [17]. An early investigation revealed the high sensitivity of fish to FEN [18]; FEN can accumulate in fish and is highly toxic to them [13]. However, there are only limited reports about the harmful effects of FEN on fish: FEN could induce sublethal effects on grass carp (*Ctenopharyngodon idella*) [19], alter biochemical indexes in mossul bleak (*Alburnus mossulensis*) [14], affect the embryonic development of zebrafish (*Danio rerio*) [16,20], and induce neurotoxicity in *Danio rerio* and common carp (*Cyprinus carpio* L.) [15,21], the possible changes in the body and the potential mechanisms caused by FEN have not been adequately elucidated.

Common carp (*Cyprinus carpio* L.), a commercially significant fish species, is extensively cultivated in freshwater environments worldwide. It represents one of the most prevalent aquatic animals exposed to various water contaminants and is frequently employed for conducting research on aquatic toxicology [22,23]. Therefore, in this study, carp were exposed to environmentally relevant levels of FEN for 14 d to investigate changes in their bodies and the underlying mechanism. The present results will contribute to a comprehensive understanding of the potential harmful impacts of FEN on fish and provide insights for risk assessment of FEN in relation to fish.

## 2. Results

### 2.1. Serum Biochemical Parameters

Compared to the control groups, the GLU contents in the carp serum following FEN exposure were noticeably increased at 7 (exception of 0.45 μg/L FEN treated) and 14 d (Figure 1). The TG and T-CHO levels were markedly inhibited in both treatment groups at 7 d (except for T-CHO in 0.45 μg/L FEN-treated groups), but generally upregulated at 14 d, although they did not reach statistical significance (except for TG contents in 1.35 μg/L FEN-treated groups) (Figure 1). Additionally, the ALB levels were generally decreased, especially after a 14 d treatment with 1.35 μg/L FEN, which showed significant differences compared to the control groups (Figure 1). The activities of ALT and AST in serum were generally elevated after being FEN treated for 7 and 14 d, and particularly after 1.35 μg/L FEN treated (Figure 1). After treatment with FEN at a concentration of 0.45 μg/L, ALP levels were noticeably elevated at 7 d, and, after 1.35 μg/L FEN exposure, ALP activities were noticeably elevated at 14 d (Figure 1). Meanwhile, there were significantly decreased levels of SOD after a 7 d exposure to 1.35 μg/L FEN (Figure 1). Moreover, exposure to 0.45 μg/L FEN noticeably increased serum CAT activities at 7 d but significantly decreased them at 14 d, and there were no discernible changes in CAT activities at 7 and 14 d following 1.35 μg/L FEN exposure (Figure 1).

### 2.2. Histopahology

Compared to the controls, FEN exposure resulted in hepatic sinusoidal congestion, hepatic sinusoidal dilatation, and inflammatory cell infiltration (Figure 2A–C). Meanwhile, the renal tissue of both exposure groups showed eosinophilic cytoplasm loss, expansion of Bowman’s space, blocked renal tubules, renal tubules intertubular congestion, and aggregated melanomacrophages, especially following 1.35 μg/L FEN exposure. There was a remarkable difference with respect to the controls (Figure 2D–F).

### 2.3. Oxidative Stress-Related Parameters in Fish Liver and Kidney

In the fish liver, there was a significant rise in ROS levels at 7 and 14 d after FEN treatment (Figure 3A). The MDA contents were generally changed after FEN treatment, but this change failed to reach statistical significance (Figure 3B). In the kidney, 0.45 μg/L FEN exposure did not result in a significant alteration in ROS levels for 7 and 14 d whilst the ROS levels were noticeably upregulated at 14 d following 1.35 μg/L FEN exposure with respect to the controls (Figure 3C). Meanwhile, FEN treatment let to a significant alteration in MDA levels for 7 and 14 d (except for the groups treated with 1.35 μg/L FEN at 7 d), when compared to those of the controls (Figure 3D).

### 2.4. Inflammation-Related Parameters in Fish Liver and Kidney

After being treated with FEN at concentration of 0.45 μg/L, the TNF-α content in the liver did not show significant changes at 7 d, while it was significantly promoted at 14 d compared to the control groups. Furthermore, following treatment with FEN at a concentration of 1.35 μg/L, the levels of TNF-α in fish liver showed a significant increase at both 7 and 14 d (Figure 4A). Meanwhile, the changes of IL-1β level had similar trends to TNF-α after FEN treatment (Figure 4B). In the kidney, the TNF-α contents were remarkably upregulated in the groups treated with 0.45 μg/L FEN for a period of 7 and 14 d, while no observable changes were observed in the groups treated with 1.35 μg/L FEN compared to the control groups (Figure 4C). Meanwhile, after FEN treatment, the IL-1β levels were increased but not significant at 7 d, and remarkably improved at 14 d (Figure 4D).

### 2.5. DEGs and qPCR Validation

A total of 1494 (149 increased and 1345 decreased) and 1550 (151 increased and 1399 decreased) DEGs were exhibited in fish liver treated with 0.45 and 1.35 μg/L FEN, respectively (Appendix A), and there were 1069 overlapped DEGs in both FEN exposure groups (Appendix A). Meanwhile, fish kidney showed 1730 (316 upregulation and 1414 downregulation) and 1669 (325 increased and 1344 decreased) DEGs in the groups treated with 0.45 and 1.35 μg/L FEN, respectively (Appendix A), and there were 1114 DEGs overlapping in both FEN-treated groups (Appendix A).

The expression levels of each ten selected DEGs from liver and kidney (Appendix A) were measured by qPCR (Figure 5A,B), and the correlation analyses all revealed a strong association between qPCR and RNA-seq data (Figure 5C,D).

### 2.6. Gene Ontology (GO) Analysis of DEGs

GO analysis results revealed 436 significantly enriched terms in the liver following treatment with FEN at a concentration of 0.45 μg/L, comprising 242 biological processes (BPs), 95 cellular components (CCs), and 99 molecular functions (MFs). Following treatment with 1.35 μg/L FEN, a total of 494 terms were significantly enriched, including 282 BPs, 109 CCs, and 103 MFs. The top 30 markedly enriched terms for the two FEN-exposure groups are shown in Appendix A. Meanwhile, the DEGs were classified into 373 (211 BPs, 74 CCs, and 88 MFs) and 362 (218 BPs, 48 CCs, and 96 MF) remarkably enriched terms in the fish kidney treated with 0.45 and 1.35 μg/L, respectively. The top 30 remarkably enriched terms in the kidney after FEN exposure are shown in Appendix A.

### 2.7. Kyoto Encyclopedia of Genes and Genomes (KEGG) and Protein-Protein Interaction (PPI) Analysis of DEGs

The KEGG enrichment analysis results showed that there were 21 overlapping pathways between the liver samples treated with 0.45 and 1.35 μg/L FEN (Figure 5E, Appendix A). Additionally, the DEGs of the FEN treatment groups in the liver exhibited significant enrichment in downregulated pathways, including DNA replication, cell cycle, mismatch repair, nucleotide excision repair, and p53 signaling pathway, et al. On the other hand, drug metabolism-cytochrome P450 and metabolism of xenobiotics by cytochrome P450 were generally upregulated (Figure 5E). Meanwhile, PPI interaction network analysis results showed that the DEGs in these pathways, including *exo1*, *cdc20*, *mcm5*, *mcm2*, *pcna*, *pold1*, *ccnb1*, *ccna2*, *rfc3,* and *bub1*, which are primarily involved in liver DNA replication, cell cycle, and mismatch repair (Figure 5G, Appendix A), exhibited a significant decrease in expression following exposure to FEN (Appendix A). Meanwhile, there were 19 and 24 significantly enriched KEGG pathways in the kidney following treatment with 0.45 and 1.35 μg/L FEN, respectively (Appendix A), in which there were 16 overlapped pathways observed between the two FEN-treated groups (Figure 5F). The DEGs in the kidney were predominantly enriched in downregulated pathways, similar to those observed in the liver. The unique DEGs were enriched in largely upregulated pathways in the 1.35 μg/L groups, such as glutathione metabolism, retinol metabolism, ether lipid metabolism, PPAR signaling pathway, and fatty acid degradation, et al. (Figure 5F). PPI interaction network analysis showed that the DEGs in these pathways, including *ccna1*, *ccnb1*, *mcm4*, *chek2*, *pole*, *nip7*, and *wee2* (Figure 5H, Appendix A), exhibited a significant decrease in expression upon exposure to FEN and the top hub genes exhibited significant enrichment in the pathways associated with cell cycle, DNA replication, and p53 signaling (Appendix A).

## 3. Discussion

Serum biochemical parameters serve as crucial indices for assessing toxic effects of contaminants in fish, and variations in these indicators and enzymatic activities potentially indicate biological dysfunction and cellular or organ injury [24]. GLU is the major high-energy complex in vertebrates, and excess GLU is stored as glycogen in liver and muscle tissues, while its levels in serum can change under various stress factors which may indicate metabolic alterations or even damage of vital organs [25]. In the current study, after FEN treatment, common carp exhibited a significant upregulation in serum GLU levels (Figure 1), which was also detected in the serum of jundiá (*Rhamdia quelen*) after cypermethrin exposure [26]. This might be due to the mobilization of liver and muscle glycogen through glycogenesis, facilitating the fulfillment of increased energy demands under FEN stress, eventually causing elevated GLU levels in serum. The primary role of TG is to store and provide cellular energy, which can compensate for energy needs under stress conditions. The serum TG levels were significantly decreased after 7 d of FEN exposure but increased at 14 d (Figure 1). The decreased in serum TG may be attributed to compensation of energy demands through gluconeogenesis from non-glycogen sources, while the increase in serum TG might be due to the alterations of lipid metabolism and adipogenesis in fish caused by FEN or could result from renal dysfunction and stress [27]. Meanwhile, we speculate that the elevated TG levels may be an important reason for high levels of serum GLU in common carp after FEN exposure, as glycerol present in TG can be converted into GLU [28].

Cholesterol, primarily synthesized by the liver, constitutes an important structural component of bile acids, plasma lipoproteins, and cell membranes, and also acts as a precursor for the synthesis of all steroid hormones [29]. In this study, after FEN treatment, the contents of T-CHO were generally decreased at 7 d, particularly following exposure to 1.35 μg/L FEN, and significant differences were observed compared to the control; however, there was no significant change at 14 d (Figure 1). The decrease in T-CHO observed under FEN stress at 7 d may be attributed to impaired hepatic cholesterol biosynthesis, the utilization of T-CHO for steroid hormone synthesis, or as a consequence of gluconeogenesis [30], and the recovery of T-CHO at 14 d may be an organismal adaptation under the long-term stress of FEN. Serum ALB is a hepatic-originated protein that regulates the osmotic balance between circulating blood and tissue membrane. It functions as a carrier for xenobiotics and hormones, as well as serving as an amino acid source utilized for cellular energy requirements or secreted into the extracellular pool of amino acids [31]. In this research, the ALB contents in the fish serum generally exhibited a decrease following FEN treatment, particularly after a 14 d treatment with 1.35 μg/L FEN (Figure 1). A reduction in ALB levels was observed in common carp after treatment with another pyrethroid, λ cyhalothrin. The similar outcomes [32] exhibited suggest that the reduction in ALB levels may be due to the inhibitory effects of FEN on hepatic ALB synthesis, or the need to meet immediate energy demands for recovery from toxic stress or liver dysfunction.

ALT and AST play pivotal roles in protein and amino acid metabolism, and are generally used as crucial marker enzymes in serum to assess liver function. Elevated serum levels of ALT and AST indicate potential liver injury or disease, as these enzymes are predominantly hepatocellular and are released into the bloodstream when liver cells are damaged [33]. The serum ALT and AST activities in carp treated with FEN exhibited a general increase in the current study (Figure 1), which was also detected in the common carp serum following esfenvalerate (a pyrethroid analogous to FEN) treatment [34], indicating that FEN treatment may cause hepatocyte damage or increase cell membrane permeability, resulting in the leakage of these enzymes from cells into the serum, which could also harm metabolism of common carp [35]. ALP is an enzyme that plays a pivotal role in various biological processes encompassing membrane transport, detoxification, and phosphate hydrolysis, which can also serve as a robust indicator of liver disease [36]. Previous studies have shown that serum ALP activity in jundiá (*Rhamdia quelen*) was significantly increased after cypermethrin exposure [26] and elevated in common carp after esfenvalerate treatment [34]. A similar type of increased levels of serum ALP was also found in common carp after treatment with FEN in the present study (Figure 1), indicating that FEN exposure causes liver damage or dysfunction in fish. Taken together, serum GLU, TG, T-CHO, ALB, ALT, AST, and ALP were disrupted in FEN treatment groups, indicating hepatotoxicity and nephrotoxicity in common carp caused by FEN exposure.

To validate our speculation, histological analysis was conducted to examine the structural alterations in fish liver and kidney tissues after 14 d of FEN treatment, because the histological alterations serve as crucial biomarkers frequently utilized in toxicological research, while histopathological examination can clearly illustrate the tissue modifications and damage induced by contaminant stress [37]. In this study, it is abundantly obvious that FEN treatment altered the pathology of the fish’s liver and kidney (Figure 2), which is consistent with the findings of research on common carp treatment with another pyrethroid, cypermethrin [38], suggesting that FEN may potentially induce hepatic and renal impairments, thereby affecting their respective functions.

The current study demonstrated histopathological changes in the liver and kidney of fish after FEN treatment (Figure 2), which may be attributed to a gradual increase in ROS levels and enhanced lipid peroxidation within the tissues under FEN stress [39]. According to previous reports, FEN exposure induces excessive production of ROS and leads to oxidative stress in multiple organs of fish, and oxidative stress is regarded as a crucial toxic mechanism of FEN towards non-target organisms [40]. SOD and CAT work as first-line antioxidant enzymes, playing crucial roles in maintaining the homeostasis of ROS and safeguarding organisms against oxidative injury, in which SOD primarily facilitates the conversion of superoxide radicals (O^2•−^) into H_2_O_2_, and then CAT transforms H_2_O_2_ into oxygen (O_2_) and water (H_2_O) [41,42]. In this research, the serum SOD activities were noticeably decreased at 7 d following 1.35 μg/L FEN treatment, while no noticeable alteration was observed at 14 d after FEN treatment. While the serum CAT activities were markedly promoted at 7 d and inhibited at 14 d following treatment with 0.45 μg/L FEN, there were no noticeable changes at 14 d in the groups treated with 1.35 μg/L FEN (Figure 1). The decreased SOD activity may be related to the FEN-induced overproduction of O^2•−^, which requires the consumption of a large amount of SOD to catalyze O^2•−^ to H_2_O_2_. The elevated CAT activities may be due to the enzymatic breakdown of H_2_O_2_ into O_2_ and H_2_O, which serves as a protective mechanism against oxidative stress in cells. Conversely, the decreased CAT activity may be due to the overproduction of H_2_O_2_, leading to excessive consumption of CAT. These findings suggest that FEN treatment may disrupt the oxidation-antioxidant system in common carp. Further research revealed that exposure to FEN (0.45 and 1.35 μg/L) for durations of 7 and 14 d significantly increased the levels of ROS in the liver of fish (Figure 3A). Additionally, ROS levels were prominently upregulated in the kidney of fish after a 14 d treatment of 1.35 μg/L FEN (Figure 3C), which was also observed in the intestine of common carp after FEN treatment [43]. These findings indicate that FEN exposure may disrupt redox homeostasis in the liver and kidney of carp, and the excessive accumulation of ROS may attack cell membranes and organelle lipids, resulting in oxidative damage to these organs. This is supported by elevated MDA levels (Figure 3B,D), since MDA is a product of lipid peroxidation and considered a crucial indicator of oxidative stress due to its association with cellular damage mechanisms [44].

Previous research has shown that the pyrethroid pesticide λ-cyhalothrin can induce increases in the contents of TNF-α and IL-1β, thereby stimulating the generation of free radicals and subsequently leading to oxidative injury [45]. In this study, the levels of TNF-α and IL-1β were generally elevated in both fish liver and kidney following FEN treatment (Figure 4A–D), suggesting their potential involvement in mediating FEN-induced oxidative damage in fish liver and kidney [45], although the specific mechanism requires further investigation. TNF-α and IL-1β are frequently considered pivotal indicators for inflammatory reactions [46]. Therefore, the elevated levels of TNF-α and IL-1β observed in the liver and kidney of FEN-treated fish suggest the occurrence of inflammation, which is further supported by the H&E staining results revealing the inflammatory cell infiltration occurred in these tissues (Figure 2).

Given the significant impacts of FEN exposure on fish liver and kidney, a transcriptome technique was used to further explore the specific biological processes and molecular mechanisms under conditions of FEN exposure. In this study, the carp liver exhibited a total of 1494 and 1550 DEGs, while the fish kidney showed 1730 and 1669 DEGs in the groups treated with 0.45 and 1.35 μg/L FEN, respectively (Appendix A), and the qPCR and RNA-seq analysis results exhibited a significant linear correlation (Figure 5C,D), confirming that the transcriptional analysis results were reliable, revealing a diverse range of transcriptional changes in the carp liver and kidney after FEN treatment, potentially impacting the biological functions in these organs. GO analysis results revealed that DEGs in the liver and kidney of all FEN-treatment fish were related to many aspects of CC, BP, and MF (Appendix A), suggesting that FEN induces many negative impacts and even interferes with normal physiological function in fish liver and kidney.

The KEGG analysis results revealed a significant enrichment of DEGs in liver and kidney across multiple pathways (Figure 5E,F, Appendix A). Interestingly, after FEN exposure, the most significant enrichment of DEGs in both liver and kidney was in the cell cycle signaling pathway (Figure 5E–H). Cell cycle is the basic process of cell life activities, which is tightly regulated in a precise sequential manner by the cell cycle proteins (Cycs) and complexes formed by cell cycle protein-dependent kinase (CDKs) [47]. Aberrant alterations of cell cycle-related genes may impact cell fate, such as proliferation, differentiation, aging, and death. Prior research has demonstrated that parts of the Cycs and CDKs genes were significantly inhibited in ticlopidine-treated zebrafish larvae, indicating that ticlopidine could affect cell proliferation and subsequently induce hepatotoxicity [48]. In this research, the genes related to Cycs (such as *ccna1*, *ccnb1*, *ccnb2*, *ccnd2* and *ccne1*) and CDKs (such as *cdk6*, *cdk7*, *cdkn1b* and *cdkn1c*) were downregulated in the liver and kidney of fish after FEN exposure (Figure 5G,H and Appendix A). This indicates that FEN may exert inhibitory effects on the proliferation of hepatocytes and renal cells by downregulating cell cycle-related genes, thereby potentially leading to hepatotoxicity and nephrotoxicity. Furthermore, the DEGs in liver and kidney were significantly concentrated in the DNA replication pathway (Figure 5E–H). DNA replication, which occurs during the S phase of the cell cycle progression, is considered to regulate cellular growth in conjunction with the cell cycle pathway [49]. In this study, the transcriptomic results of the fish liver and kidney after FEN exposure showed down-regulation of DNA replication-related genes, containing the DNA replication licensing factor MCM complex (such as *mmcm3*, *mcm4,* and *mmcm6*), DNA replication factor C complex (such as *rcf2*, *rcf3*, *rcf4,* and *rcf5*) and origin recognition complex (such as *orc1*, *orc3*, *orc4,* and *orc6*) (Figure 5G,H and Appendix A). Moreover, the GO analysis results showed that the MCM complex was the main significant category of the cellular component (Appendix A), which is required for cell cycle and DNA replication [50], suggesting that inhibition of DNA replication occurs in the carp liver and kidney under FEN stress.

Meanwhile, the DEGs in the liver and kidney of common carp treated with FEN significantly enriched KEGG pathways including nucleotide excision repair, mismatch repair, and p53 signaling pathway; notably, a majority of these DEGs exhibited downregulation (Figure 5E,F). Mismatch repair proteins contribute to maintaining genome stability by recognizing and facilitating an appropriate cellular response to abnormal DNA structures [51]. The nucleotide excision repair system represents the primary defense mechanism for rectifying DNA injury [52], while p53 plays a vital role in maintaining genome stability and regulating DNA damage repair [53,54]. Therefore, we speculate that FEN may impair the process of DNA replication and cellular repair function, potentially inducing DNA injury to the liver and kidney of carp. In this research, the KEGG results also revealed that the metabolism of xenobiotics by cytochrome P450 pathway was significantly enriched by upregulated DEGs in the carp liver following FEN treatment (Figure 5E), indicating the activation of this pathway, similar to previous reports that revealed that xenobiotics exposure can induce this pathway in fish [55,56,57]. This may be related to the function of the liver as an extremely crucial metabolic organ, participating in the biotransformation and catabolism of FEN. In the kidney, particularly following treatment with 1.35 μg/L FEN, there was a significant enrichment of DEGs within multiple pathways associated with the endocrine system, lipid metabolism, amino acid metabolism, carbohydrate metabolism, and metabolism of cofactors and vitamins (Figure 5F). This may be related to the crucial role that fish kidneys play in maintaining homeostasis of body fluids and excreting toxic xenobiotics. The differential response to FEN toxicity observed in the liver and kidney may be attributed to their distinct physiological functions, although further studies are needed.

## 4. Materials and Methods

### 4.1. FEN and Common Carp

The FEN compound (purity ≥ 99.0%) was procured from Sigma (Shanghai, China) and subsequently dissolved in dimethyl sulfoxide (DMSO, Sigma, Shanghai, China) to prepare stock solutions, which were stored at 4 °C prior to use. The common carp (12.24 ± 2.14 g) were procured from an aquaculture farm located in Henan, China, and temporarily reared within a recirculating aquaculture system in our laboratory (temperature, 26.0 ± 1.0 °C; light:dark photoperiod, 14 h:10 h). The carp were fed with commercial food twice daily, and no fish died during the period.

### 4.2. Experimental Design and Sample Collection

Briefly, healthy carp were selected and evenly allocated into three groups, and each group contained 17 fish (n = 3 replicates/group). The concentrations of FEN selected for each group were 0, 0.45, and 1.35 μg/L (with each group containing 0.0135‰ DMSO), respectively, which corresponded to environmentally relevant levels, as described in the introduction section, with the aim to investigate potential adverse impacts on fish. The fish were fed twice daily at a temperature of 26.0 ± 1.0 °C, maintaining a light–dark photoperiod of 14 h:10 h, following the guidelines outlined in OECD 204 [58]. Additionally, three-quarters of the treated solution was renewed daily. Each experimental group consisted of three replicates. The experiment lasted for 14 d, and no fish mortality occurred during the exposure period. All procedures were conducted strictly in accordance with approval from the Ethics Committee of Henan Normal University.

The water was collected every three days before renewing the exposure solution in each group, and FEN quantification was performed using HPLC [59]. The measured concentrations of actual FEN in the water samples were 0.4126 ± 0.0023 and 0.4025 ± 0.0130 μg/L at 7 and 14 d, respectively, in the groups exposed to a concentration of 0.45 μg/L FEN; and 1.3506 ± 0.0443 and 1.4317 ± 0.0371 μg/L at 7 and 14 d, respectively, in the treatment groups exposed to a concentration of 1.35 μg/L FEN.

Following 7 and 14 d of treatment, six fish were selected from each group at each time. Blood samples were collected from the caudal vein of fish and immediately dissected on ice. The carp liver and kidney samples were stored at −80 °C for further analysis. On day 14, five fish from each treatment group were chosen for hematoxylin-eosin (H&E) staining of their liver and kidney tissues.

### 4.3. Biochemical Parameters Examination

Pretreatment of the carp liver and kidney samples was carried out, with detection of levels of ALT (# C009-2-1), AST (# C010-2-1), ALP (# A059-2), SOD (# A001-3-2), CAT (# A007-1-1), GLU (# A154-1-1), ALB (# A028-2-1), TG (# A110-1-1), T-CHO (# A111-1-1), and MDA (# A003-1-2) (Nanjing Jiancheng, Nanjing, China), and ROS (# E-335577), TNF-α (# E-43904), and IL-1β (# E-43902) (Andygene, Beijing, China). The operation procedures can be found in Appendix A.

### 4.4. Transcriptomic Analysis

The fish liver and kidney tissues underwent high-throughput transcriptomic sequencing, with the RNA-seq analysis accompanied by comprehensive details provided in Appendix A.

### 4.5. Quantitative PCR (qPCR) Analysis

Total RNA was extracted from the liver and kidney, cDNA synthesis was performed, and qPCR reactions were carried out as previously described [41]. Experimental details and primer sequences (Appendix A) can be found in Appendix A.

### 4.6. Histological Analysis

Fish liver and kidney samples were fixed in 4% paraformaldehyde for up to 24 h, followed by dehydration and paraffin coating. They were cut into slices measuring 5–10 μm in thickness, which underwent staining with H&E, and observed using an optical microscope, following the established protocol [60].

### 4.7. Statistical Analysis

The data analysis was conducted using SPSS 23.0 software (Chicago, IL, USA). The normal distribution and homogeneity of variance were assessed using Shapiro–Wilk’s and Levene’s tests, respectively. Significant differences (*p* < 0.05) between the FEN exposure groups and control groups were examined through one-way analyses of variance (ANOVA), followed by post hoc pairwise comparisons based on Dunnett’s test.

## 5. Conclusions

Taken together, our results showed that FEN exposure induced hepatotoxicity and nephrotoxicity in common carp, partly by altering the serum biochemical parameters, causing oxidative stress and inflammatory response, and impeding the proliferation of hepatocytes and renal cells, leading to dysfunction of liver and kidney, which then impacts the health of common carp (Figure 6). Overall, the present findings enhance our comprehension of the hepatotoxicity and nephrotoxicity of FEN and provide insights for risk assessment of FEN in fish.

## Figures and Tables

**Figure 1 ijms-25-09822-f001:**
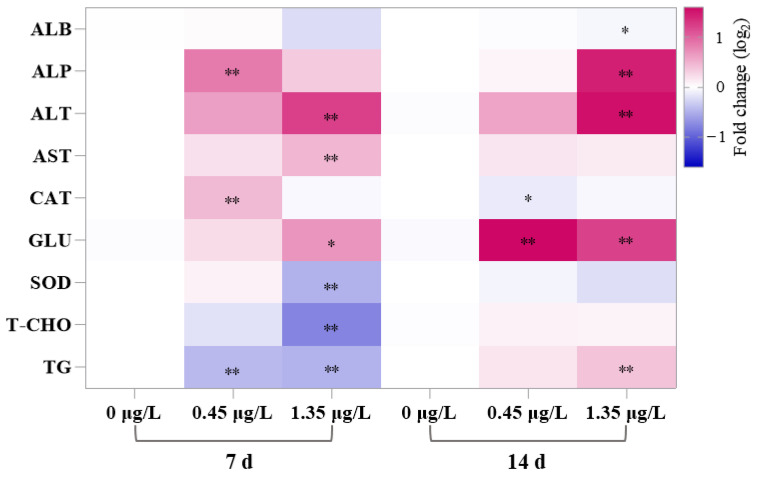
Changes of the serum biochemical indicators in common carp after a 14 d exposure to FEN. Values are presented as the mean ± SD. * *p* < 0.05 and ** *p* < 0.01 compared to the control.

**Figure 2 ijms-25-09822-f002:**
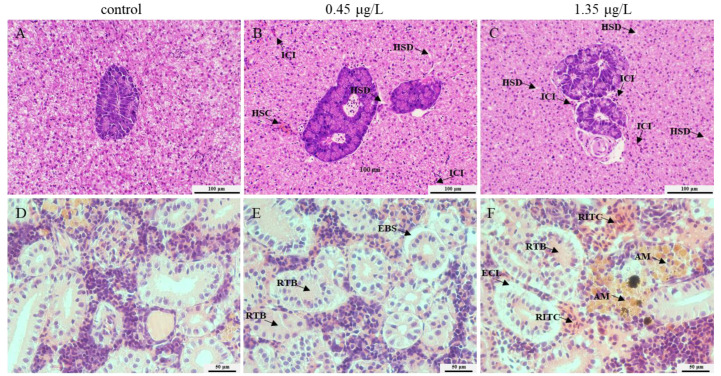
Histopathological analysis of the carp liver and kidney after FEN treatment. Representative H&E-stained liver graphs from controls (**A**), 0.45 μg/L (**B**), and 1.35 μg/L groups (**C**). Representative kidney images from control (**D**), 0.45 μg/L (**E**), and 1.35 μg/L groups (**F**). AM: aggregated melanomacrophages; EBS: expansion of Bowman’s space; ECL: eosinophilic cytoplasm loss; HSC: hepatic sinusoidal congestion; HSD: hepatic sinusoidal dilatation; ICI: inflammatory cell infiltration; RTB: renal tubules blocked; RTIC: renal tubules intratubular congestion. Black arrows indicate the site of tissue injury. (Scale: 100 μm; 50 μm).

**Figure 3 ijms-25-09822-f003:**
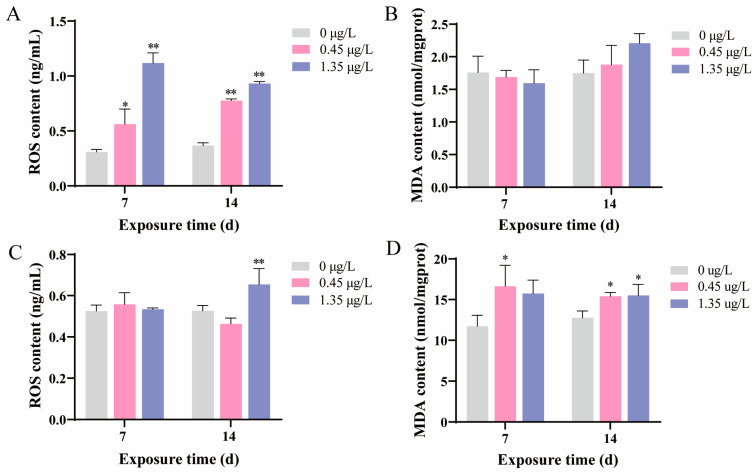
Changes of oxidative stress-related indicators in the carp liver and kidney after a 14 d exposure to FEN. Values are presented as the mean ± SD. * *p* < 0.05 and ** *p* < 0.01 compared to the controls. d represents the term day. (**A**) ROS levels in the carp liver. (**B**) MDA contents in the carp liver. (**C**) ROS levels in the carp kidney. (**D**) MDA contents in the carp kidney.

**Figure 4 ijms-25-09822-f004:**
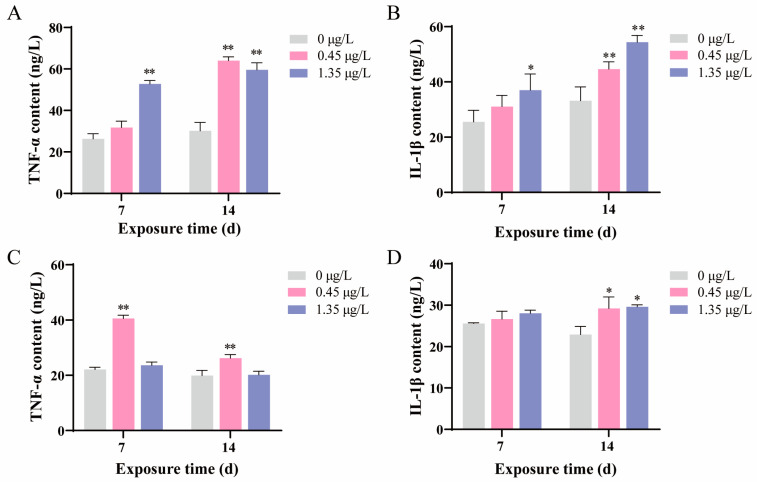
Changes of inflammation-related indicators in the carp liver and kidney after a 14 d exposure to FEN. Values are presented as the mean ± SD. The presence of asterisks indicates significant differences as compared to the control groups (* *p* < 0.05, ** *p* < 0.01). d represents the term day. (**A**) The levels of TNF-α in the liver. (**B**) IL-1β contents in the liver. (**C**) TNF-α levels in the kidney. (**D**) IL-1β contents in the kidney.

**Figure 5 ijms-25-09822-f005:**
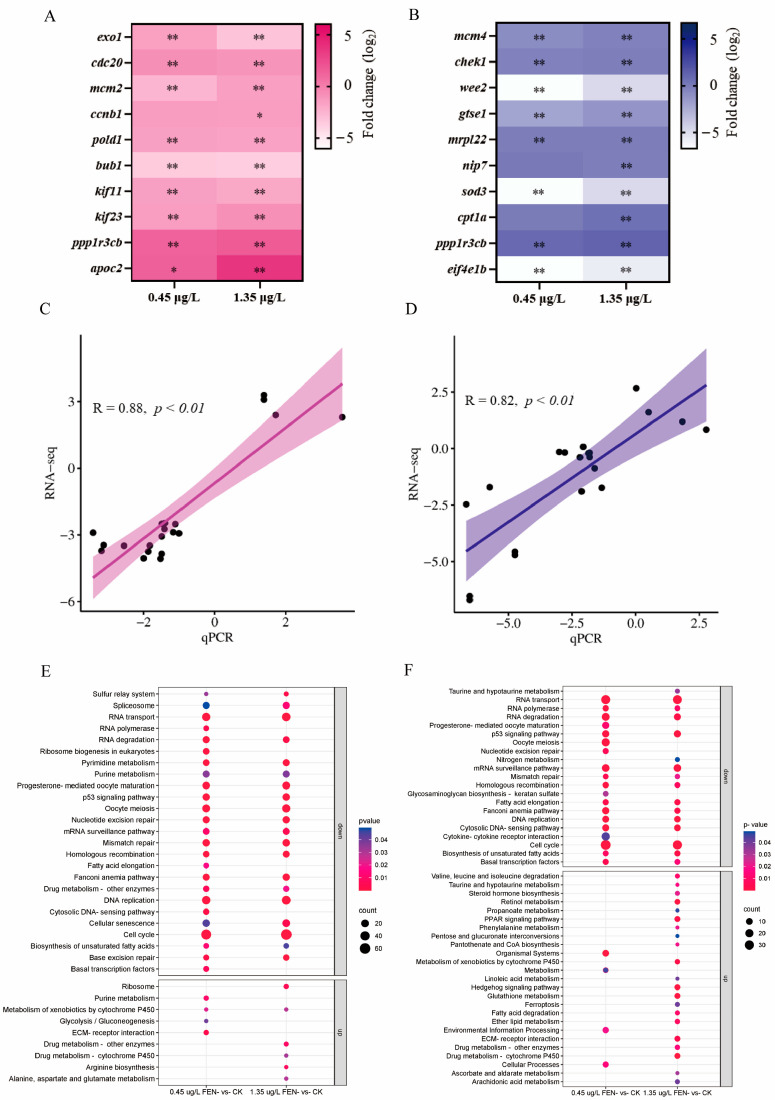
Transcriptional changes in the liver and kidney of common carp after a 14 d exposure to FEN. Gene expression detection in the carp liver (**A**) and kidney (**B**) after FEN treatment using qPCR. Correlative analyses of qPCR and RNA-seq analysis results in the carp liver (**C**) and kidney (**D**). KEGG pathways of DEGs in the liver (**E**) and kidney (**F**) in both FEN-treated groups. Graph of significant gene networks associated with DNA replication and cell cycle in the carp liver (**G**) and kidney (**H**). * *p* < 0.05 and ** *p* < 0.01 compared to the controls.

**Figure 6 ijms-25-09822-f006:**
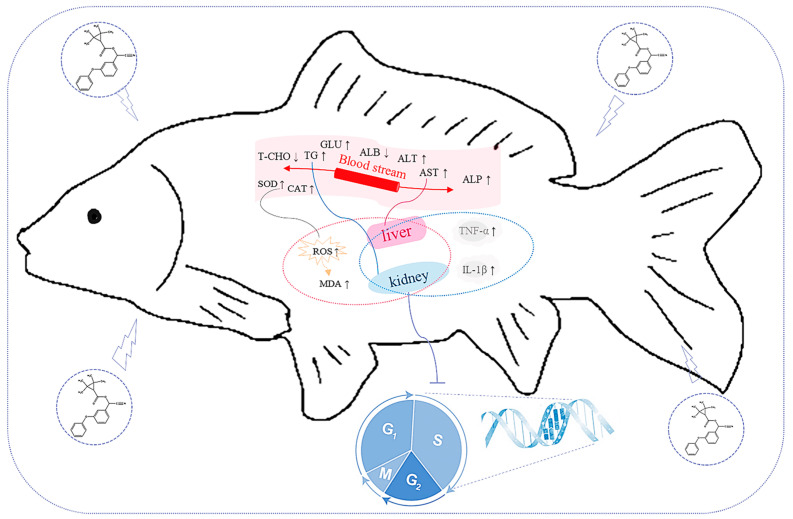
A summary of hepatotoxicity and nephrotoxicity of FEN in common carp. Arrows indicate up-regulation (↑) or down-regulation (↓) of expression.

## Data Availability

Data supporting the reported results are contained within the article.

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
