# Peer review of "Risk Assessment of Fenpropathrin: Cause Hepatotoxicity and Nephrotoxicity in Common Carp (Cyprinus carpio L.)"

_ijms, 2024, doi:10.3390/ijms25189822_

Round 1

Reviewer 1 Report

Comments and Suggestions for Authors

The article entitled 'Risk assessment of fenpropathrin: cause hepatotoxicity and nephrotoxicity in common carp (Cyprinus carpio L.)' is interesting and was well written. Before accepting it for publication in a journal, I would suggest making a few minor changes, which I list below:

lines 68-70 - please add English names of fish species

lines 73-74 - Is it only in Asia that carp is often bred in aquaculture? In many European countries it is the most important species bred in inland aquaculture. Please complete this section.

Figure 2 - please bold the scale bars to make it more visible

Figure 3 and 4 - please check that the letter designations are properly assigned to the individual graphs (they have shifted in the pdf)

Lines 223 and 267 - please separate the common names from the scientific name. Only the Latin name should be written in italics and in brackets

lines 406 - 416 - please add specific information on how many fish were in the experimental group

Reviewer 2 Report

Comments and Suggestions for Authors

Review of the Manuscript Number IJMS-3187658 entitled: “

 Risk assessment of fenpropathrin: cause hepatotoxicity and ne-2 phrotoxicity in common carp (Cyprinus carpio L.)” by  Gongming Zhu, Zhihui Liu, Hao Wang1, Shaoyu Mou, Yuanyuan Li, Junguo Ma and Xiaoyu Li.

 Pyrethroid insecticides are widely used worldwide. Fenpropathrin (FEN) is a new synthetic, highly effective, broad-spectrum pyrethroid acaricide and insecticide that is used, for example, to protect fruit, vegetable, cotton, cereal crops.

Currently, the presence of FEN has been frequently discovered in biota and environmental samples. Fish are important aquatic organism and exhibit a high degree of sensitivity to changes in the aquatic environment, and frequently used as an important bio-indicator species for aquatic pollution.

These studies are important for understanding the possible harmful effects of FEN on fish, especially regarding liver and kidney damage. New information allows for a better assessment of the risk of FEN on fish.

I consider the results obtained at work as valuable.

To help you improve the manuscript, I would like to propose:

Figure 3 – could you explain what does “d” mean?

Please explain the abbreviations – DEGs, KEGG, PPI

Are the conclusions together with discussion? I suggest separating them.

According to the authors, can the obtained studies be used to build a model for assessing the toxicity of other fish species?

Please also provide more information on similar studies for other pesticides.

Check the whole manuscript, please. Authors should carefully check for grammar, punctuation and sentence structure before submitting the revised paper.

      It is recommended that this manuscript can be accepted for publication after minor revision.

Dół formularza
